# Evaluation of Age-Dependent Changes in the Coloration of Male Killifish Nothobranchius Guentheri Using New Photoprocessing Methods

**DOI:** 10.3390/biology11020205

**Published:** 2022-01-27

**Authors:** Dmitry L. Nikiforov-Nikishin, Nikita I. Kochetkov, Ekaterina V. Mikodina, Alexei L. Nikiforov-Nikishin, Yuri G. Simakov, Natalya A. Golovacheva, Alexander V. Gorbunov, Sergei N. Chebotarev, Evgeniya Yu. Kirichenko, Igor Yu. Zabiyaka, Ivan S. Pastukhov, Anzhelika B. Bren

**Affiliations:** 1Department of Biotechnology and Fisheries, Moscow State University of Technologies and Management (FCU), 73, Zemlyanoy Val Str., 109004 Moscow, Russia; niknikdl@rambler.ru (D.L.N.-N.); 9150699@mail.ru (A.L.N.-N.); usimakov@yandex.ru (Y.G.S.); molekula00@inbox.ru (N.A.G.); akvabiotex@rambler.ru (A.V.G.); ivanpeskar@gmail.com (I.S.P.); 2Russian Federal Research Institute of Fisheries and Oceanography (VNIRO), Verkhnyaya Krasnoselskaya St., 17, 107140 Moscow, Russia; arizon99@rambler.ru; 3Management Department, Moscow State University of Technologies and Management (FCU), 73, Zemlyanoy Val Str., 109004 Moscow, Russia; Chebotarev.sergei@gmail.com; 4Faculty “Bioengineering and Veterinary Medicine”, Don State Technical University, Russian Federation, 1 Gagarin Sq., 344000 Rostov-on-Don, Russia; kler.smo@gmail.com; 5Department of Physical and Applied Material Science, Don State Technical University, Russia Federation, 1 Gagarin Sq., 344000 Rostov-on-Don, Russia; zabiyakaigor@gmail.com; 6Academy of Biology and Biotechnology, Southern Federal University, 344090 Rostov-on-Don, Russia; brenanzhelika@yandex.ru

**Keywords:** coloration, *Nothobranchius guentheri*, age-dependent changes, live fish photography, visualization of morphological characteristics

## Abstract

**Simple Summary:**

This paper proposes a new methodology for evaluating fish coloration, which allows us to identify differences in the intensity of coloration of specific areas of the body. Changes in fish coloration occur during growth and under the influence of environmental factors. Male fish belonging to the family *Nothobranchius* are characterized by extremely diverse coloration, depending on the age of the fish, environmental factors, and social hierarchical status. As the lifespan of this genus of fish is very short (12–14 months), studies on age-dependent changes are possible. In this study, we demonstrate correlations between the coloration of particular body zones of male *Nothobranchius guentheri* and age using new photofixation methods and image processing software. This methodology can be applied to other fish with unique coloration patterns, for example, family *Cichlidae* and order *Cyprinodontiformes*.

**Abstract:**

Fish as model objects have found wide applications in biology and fundamental medicine and allow studies of behavioral and physiological responses to various environmental factors. Representatives of the genus *Nothobranchius* are one of the most convenient objects for such studies. Male fish belonging to the family *Nothobranchiidae* are characterized by extremely diverse coloration, which constantly changes, depending on the age of the fish, environmental factors, and social hierarchical status. These fish species are characterized by a short life cycle, which allows changes in coloration, an indicator of the ontogenesis stage, to be estimated. Existing methods of fish color assessments do not allow the intensity of coloration of particular body zones to be clearly differentiated. In the present study, we suggest a method of two-factor assessment of specific fish body zones using modified methods of photofixation and image processing software. We describe the protocol of the method and the results of its application to different-aged groups of male *Nothobranchius guentheri*. The coloration of selected areas (i.e., red spot on the gill cover (RSGC), black border on the caudal fin (BBCF), and white border on the dorsal fin (WBDF)) differed significantly according to the size and age of the fish (*p* < 0.05). The data obtained suggest that *N. guentheri* can be a model for studying aging by the intensity of body coloration in males.

## 1. Introduction

Fish as model objects have found wide applications in biology and fundamental medicine and allow studies of behavioral and physiological responses to various environmental factors. In particular, Zebrafish (*Danio rerio*) [1] and medaka (*Oryzias latipes*) [2] have become widely used as representative teleosts. However, this model organism is not suitable for studying the aging process or changes in coloration with age due to the longer life span. In fish, coloration features and patterns are manifested during the course of reaching sexual maturity. It is difficult to study colorations and their changes at various stages of ontogenesis due to species differences and different rates of maturation [3].

The family *Nothobranchiidae* comprises a large group of fish, which mainly inhabit shallow ephemeral pools in northern Africa. As a result, they have formed a unique mechanism of reproduction and development, where the individual stages of ontogenesis are very close to each other [4]. *Nothobranchiidae* are short-lived, relatively easy to husbandry, transgenetic methods and genome sequencing have been determined, and they are much cheaper to keep than laboratory mice [5]. As such, representatives of this family are a convenient object for studying the processes of evolution, development, and aging [6,7,8,9,10].

The diverse coloration of male fish belonging to the *Nothobranchiidae* family cannot be explained only by mechanisms of sexual selection but is most likely a consequence of diverse ecological conditions, isolation of individual populations [11,12], and heterogeneity of this group of species. The following species are most often used to study the aging process: *Nothobranchius*
*furzeri* [13], *N. rachovii* [14], and *N. guentheri* [15]. Each of these species has advantages and disadvantages in terms of studying the aging process. Among these species, *N. rachovii* has the shortest life cycle (5−6 months) but has one of the longest embryonic development durations (8−9 months), which makes its cultivation in laboratory conditions difficult. In contrast, *N. guentheri* has a longer life span and short embryonic diapause, which means that synchronized cultures can be obtained for research [8]. Representatives of the genus *Nothobranchius* are characterized by a high diversity of karyotypes, which, perhaps, plays an important role in their adaptation to unstable environmental conditions [16].

The coloration of male fish is characterized by phenotypic variability, which can vary within a wide range, both within members of different populations and among fish of the same group [11]. Thus, fish belonging to this genus appear to possess a divergent karyotype [17].

Changes in morphometric parameters of the body coloration of the genus *Nothobranchius* can provide an objective picture of maturation and aging (Figure 1).

The genus Nothobranchius is characterized by apparent sexual dimorphism and dichromatism [18]. As males of this genus are usually much brighter colored than females [19,20], they can be used for color assessments. Criteria other than color (e.g., fertility) are used to assess growth-related parameters and maturation of female fish [21]. The presence of distinct differences in coloration and body shape suggests that visual signals play an important role in the choice of a sexual partner [18,22].

In addition to age, fish coloration, particularly coloration of the genus *Nothobranchius*, can change under the influence of the following factors: in response to environmental factors [23,24,25], social position in a group of individuals [26,27], feeding regime and food availability, drags affecting the neurohumoral system (e.g., exogenous androgens) [20,28], and according to some authors (N. Papa, personal communication [29]), during the aging process.

The primary aim of the present study was to investigate coloration changes in *N. guentheri* males, depending on the age and size of the fish. An additional aim was to develop a method of their quantitative measurement based on novel photofixation/photoprocessing methods, applicable to most species of the family *Nothobranchiidae* and other fish, which are characterized by unique coloration traits. Such a method will expand the use of N. guentheri as an alternative model for biological studies. The hypothesis of the present study was that differences in the coloration of particular body zones of *N. guentheri* males would be associated with different stages of maturation and aging. This hypothesis is based on the assumption that due to the short life cycle in the ontogenesis of this fish species, there is a clear staging of age-related changes, including coloration.

## 2. Materials and Methods

### 2.1. Fish Housing

The isolate *N. guentheri* Zanzibar TAN 14-02 was obtained from the collection of the Engelhardt Institute of Molecular Biology of the Russian Academy of Sciences. In total, 400 individuals of both sexes (age range: 2−8 months) were included in this study. The fish were divided on the basis of sex and size parameters and kept in 50 L aquarium tanks (45 × 45 × 15 cm^3^, W × L × H), with constant aeration at a temperature of 22 ± 2 °C and pH of 7.2−7.6. The water in these holding tanks was changed according to the following schedule: partial (once every 2 days) and complete (once every 7 days). The fish were fed live food (*Daphnia magna*, larvae of the family *Chironomidae*, *Artemia salina*) twice a day, at 12:00 and 18:00 h.

### 2.2. Anesthesia of Fish

For photofixation, the fish were anesthetized to ensure no locomotor activity during the imaging period (4−5 min). In order to avoid stress, which induces changes in coloration [30], the fish were sedated in a solution of MS-222 (0.1 mg/L) [13], after which their physical activity significantly decreased [31,32]. Subsequently, the fish remained in the aquarium for 5 min.

After photofixation, each fish was moved to a preprepared rehabilitation tank, where, after full recovery of cognitive function [33], the fish was returned to the holding tank. In order to exclude the possibility of behavioral changes in response to the sedation procedure, the fish was not exposed to the photofixation procedure again until 5 days after the first session.

### 2.3. Imaging

In digital imaging, there were two key goals: to ensure the reliability of the color display and to ensure the repeatability of the data obtained. Previous studies provided detailed descriptions of the principles underlying fish photography [34,35]. Photofixation was conducted in accordance with the protocols outlined in the aforementioned studies, taking into account the purposes of the present research and features of the coloring of the fish species in the present study.

An example of the imaging setup is shown in Appendix A. The camera was installed in front of the aquarium tank at a distance sufficient for focusing. An artificial light source was used, taking care to ensure that the obtained camera image was clear, with no negative effects (glare, overlight, etc.). We used a studio diode ring illuminator (Raylab RL-0518 Kit; Raylab, St Petersburg, Russia), where the color temperature was set to 4200 K (neutral color range), and a Nikon D5000 camera (Minato, Tokyo, Japan).

An aquarium tank with the following dimensions was used for fish photofixation: 55 × 35 × 20 mm^3^ (Appendix A). The tank was made of quartz glass to achieve high optical transparency. The position of the tank remained unchanged during the whole period of photofixation to obtain high-quality photos. In order to compensate for artifacts and simplify further processing of the obtained image, the aquarium was set against a black background.

### 2.4. Photographic Imaging Conditions

In order to study age-dependent changes in coloration, the fish were divided into two groups based on the known age of the fish from the time of their hatching. The first group comprised individuals with a body length of 2.28–2.60 cm (2.5 months) (*n* = 24), and the second group consisted of individuals with a body length of 3.10−4.18 cm (8 months) (*n* = 36).

Photofixation of living objects to quantify coloration is a complex methodological task [36]. Photofixation protocols previously described by a number of researchers [34,37,38] were modified based on the species-specific characteristics of *N. guentheri*. The study design is presented in Appendix A. In our experiments, to obtain photographs suitable for further processing, the criteria were as follows: the fish remained motionless during the whole imaging period (see Section 2.2), the left lateral part with all morphological features of the fish [26] was visible on the image, and for reliable coloration evaluation, all the photographs of the fish were taken under identical lighting conditions (4200 K) and using the same camera settings (ISO 400−500, F 5.6, 1/60 s). Due to the strict standardization of the imaging conditions, color balance image calibration, usually used in similar studies [34,39], was not required.

Photofixation was performed according to the following protocol: male *N. guentheri* fish were moved from the housing tank to the sedation tank in groups of 10 individuals. After the onset of sedation, each fish was moved to the aquarium tank for photofixation. After the photographs were obtained, the fish (not more than 2−3 min postsedation) was moved to a specially prepared container for rehabilitation. In total, 104 male *N. guentheri* of different ages were photographed. From the database of the obtained photographs, we selected 60 of the most representative images suitable for processing (images with defects and distortions were not used).

### 2.5. Image Processing

Digital images in RAW format were processed using Adobe Photoshop ® software, using standard tools to crop the image of each fish from the original image. Image processing to obtain numerical data on coloration intensity was performed using ImageJ software (Wayne Rasband (NIH); https://imagej.nih.gov/ij/; accessed on 21 August 2021). The image processing technique was as described in previous studies [30,39,40,41]. The colors were coded according to hue, saturation, and brightness parameters [42].

In order to obtain reliable numerical data on coloration intensity, we apply a new method where fish coloration is presented as a set of reference zones, which are then evaluated according to two criteria: the coloration area and the modal (MoGV) or mean (MeGV) gray value. The basis of this method is the way in which a digital photo is represented as a set of pixels, each of which has a corresponding value defined by a color model. Once the resolution of the photographic image is sufficient, digital data can be obtained for further processing and statistical analysis. Determining the color of a particular zone is directly dependent on the resolution of the photographic image, as a single pixel may not be included in the determined zone at a low resolution [36].

The grayscale point value is a color mode in which each component of the original image is assigned a value equal to a shade of gray (from 1 to 255). In this way, the lightest/whitest pixels are assigned values close to 255, whereas the darkest/blackest pixels areas assigned values closer to zero. Applying this mode of displaying the zones under study allows us to estimate the degree of coloring intensity.

For this study, the authors identified two sets of representative coloration zones that most clearly reflect the phenotypic diversity of *N. guentheri* males and also play an important role in determining the social rank of an individual. Brightly colored body areas (zones) predetermine sexual behavior and mate choice [43] and are primally located on the lateral side of fish. In other fish species, similar zones with similar functional significance can be distinguished [43,44].

The first set of body zones are visually the most distinguishable and are as follows (Figure 2):Red spot on the gill cover (RSGC);Black border on the caudal fin (BBCF);White border on the dorsal fin (WBDF).

The names of the first set of zones are based on the subjective perception of colors and used only to describe them in the text. The WBDF and RCTPB zones were not used in this study but may be used in further research

The zones in the first group are distinguished by the fact that they appear at the earliest stages of postembryonic development (3–4 weeks post-hatching) and remain visible, even in the presence of a marked change in fish coloration in response to external factors. In our opinion, this set of zones most accurately reflects the phenotypic dispersion within the studied isolate *N. guentheri*. Phenotypic dispersion should be minimal, as fish belong to the same isolate and has a similar genetic structure.

The zones in the second group comprised a set of areas on the lateral part of the fish, matching the chosen gradation of colors. The zones used in the present study were:Zone of black zones on the dorsal and caudal fins, black region of interest (BROI) (Appendix A);Zone of red zones on the entire lateral part, red region of interest (RROI) (Appendix A);Zone of light blue areas on the entire lateral part, light blue region of interest (LBROI) (Appendix A).

These zones were selected using the threshold color tool in ImageJ, adjusting the hue, saturation, and brightness of the original image (Appendix A). Numerical processing and estimation of the values of the zones in the second group allowed us to obtain the maximum data (MoFV, MeGV) from discrete information. We believe the first and second sets of zones can be considered objective criteria to evaluate coloration intensity, as they accurately reflect coloration in N. guentheri males linked to age and social status.

### 2.6. Statistical Analysis

The data are presented as mean ± standard deviation (SD). A comparative analysis of the different study groups was performed using the Student’s t-criterion. A *p*-value of <0.05 was accepted as statistically significant. In order to determine the normality of the data distribution, the Shapiro−Wilk test was used. The correlation between different morphometric parameters was determined using Pearson’s correlation, with the Student’s t-distribution used to calculate the significance. Paired linear regression was performed for parameters with a significant value of the correlation coefficient. The correlation lines were compared using an analysis of covariance (ANCOVA). Statistical data processing was performed using GraphPad Prism version 8.0 software (GraphPad, San Diego, CA, USA).

## 3. Results

The graphs in Figure 3 and Table 1 show the measurement data for the 60 experimental fish by the studied zones and MoGV and MeGV parameters. Numerical data are presented in Table 1 and Table 2. The angle of slope of the correlation line significantly differed from zero (*p* < 0.05) in the zones: MeGV for BROI (R^2^ = 0.1512); MoGV and MeGV for LBROI, respectively (R^2^ = 0.07053 and R^2^ = 0.07935). The most significant correlation between fish size and the studied gray values was noted in the BROI zone.

For the first set of zones (BROI, LBROI, and RROI), the regression distributions of both studied gray values (MoGV and MeGV) coincided, and the slope angles did not differ significantly, resulting in a straight line. In the zones in the second group (RSGC and BBCF), we can discuss the reliable parallelism of the regression lines (*p* < 0.01) (Figure 3e,f).

In all the studied zones, the area was dependent on the size of the individual (Figure 4, Table 2). In all the studied zones, the angle of slope of the correlation line was significantly non-zero (*p* < 0.05). The smallest scatter interval and the highest correlation coefficient were noted in the following groups: RSGC (R^2^ = 0.7444) and LBROI (R^2^ = 0.723). For other coloration zones, the correlation coefficient was lower, with R^2^ = 0.05676 for the BBCF zone.

In most cases, the selected areas in each of the two studied zones showed a clear correlation with the size of the fish. Coloration area increased in accordance with the age of the fish. A small proportion of fish due to phenotypic plasticity go beyond the range of the correlation line.

For a more adequate representation of the results of the study, the male fish were divided into two groups according to size: 2.51 ± 0.11 cm (2.5 months old) and 3.26 ± 0.42 cm (8 months old) (Appendix A).

As shown in Figure 5, in the zones in the first group, there was a significant difference only in the parameter BROI between the MeGV for both age groups of fish (*p* < 0.05).

In the RSGC zone, a significant difference was noted between the MoGVs and MeGVs of the different age groups (*p* < 0.001). In the BBCF zone, the MoGV of 2.5-month-old individuals differed significantly from that of 8-month-old individuals (*p* < 0.05), as well as in the MeGV (*p* < 0.001). In the two groups of zones, there was a significant difference in coloration intensity, as shown by the MoGVs and MeGVs (*p* < 0.001). For specific coloration areas, the MeGV value was a more reliable measurement parameter than the MoGV.

Comparison of the staining areas of the studied zones (Figure 6, Table 2) showed that the intensity of coloration of the following areas differed significantly in fish of different ages: LBRIO, RROI, and RSGC (*p* < 0.05).

## 4. Discussion

Previous methods designed to study fish coloration utilized the entire fish body and spectrocolorimetry [45]. Efforts to apply these methods to assess the coloration of male *N. guentheri* did not give positive results due to issues relating to laboriousness, the complexity of obtaining serial results, and the difficulty in measuring coloration parameters in fish with a complex morphological pattern of coloration. The methods proposed earlier are better suited for the estimation of large-sized fish with homogeneous coloration [46]. In our opinion, the methods proposed in this study for evaluating the coloration of male *N. guentheri* have advantages as compared to evaluation methods used previously [34,35,37].

The development of digital photography makes it possible to obtain high-quality images, where each color zone corresponds to a large number of pixels [36]. Thus, general age- and size-related parameters, as well as the coloration of individual zones, can be assessed and compared, as demonstrated in this study.

Male fish, including male *N. guentheri*, can change the intensity of coloration depending on age, housing conditions, behavioral characteristics, and size [30]. The methodical approach proposed in this study makes it possible to standardize the color comparison in male *N. guentheri* fish.

In the zones in the first group, in the entire sample of individuals of different ages, the observed directions of linear regression can be explained by age-related changes in coloration, with brightening or darkening of the studied zones. The direction of the gray line correlation in different zones may be different (positive/negative).

Most likely, such variations in coloration are related to changes in the number of chromotocytes or pigment-protein content [47]. Changes in melanin content with age are characteristic of much bony fish, including *N. guentheri* [6]. As melanin is not the only pigment [48] involved in the formation of the coloration of the measured areas, the total value of gray varied within a slight range. Thus, in the BROI zone, age-related color brightening was clearly observed. The straight regression line for both MeGVs and MoGVs showed a negative direction. In turn, the LBROI zone had a positive regression, which points to the darkening of these zones with age. The multidirectional direction (positive in LBROI; negative BROI, BBCF) of the regression confirms the course of changes in fish coloration with increasing size and age, with these changes commencing at the time of sexual maturation and persisting almost until death [49,50]. Age-related variability can also be observed in fish with less bright coloration, although the variability is less distinct.

Some individuals differ significantly from the nominal coloration as a result of the phenotypical dispersion. The coloration of these fish cannot be interpreted within the framework applied in the present study. For example, Figure 7 shows a male *N. guentheri*, the coloration of which does not correspond to the coloration of individuals of his age and size. It can be assumed that the uncharacteristic coloration of this isolate is associated with intersex, which is observed in a percentage of individuals belonging to this family [51,52].

In the case of the zones in the first group, the differences between the MoGVs and MeGVs were minimal, as these zones of the fish body are characterized by uniform coloration, without pronounced individual differences. The stability of the coloration of these body zones may be a consequence of characteristic species or population traits that contribute to the recognition of fish in the group [53] and the choice of a mate [43,54]. The results of the present study are in accordance with those of previous research on coloration in the *Cichlidae* family [40].

The zones in the second group were characterized by a smaller coloration area. The results of the measurements of color intensity and those of the statistical analysis differed significantly in the two groups. Thus, the regression lines for the MoGVs and MeGVs did not overlap and were reliably parallel, and the mean values in both zones were almost always greater than the modal values. This may indicate that at the tissue level, pigment cells are located irregularly, causing diffuse changes in coloration [55,56]. As reported previously, pigment cells are arranged in layers, with xanthophores in the outer layer, iridophores in the middle layer, and melanophores in the inner skin layer [57,58].

The slope angle of the regression line of the RSGC zone did not differ from zero, with the gray values remaining the same during the studied lifespan of the fish. The RSGC zone is an important phenotypic trait characteristic both for this isolate and for the species as a whole. It can be assumed that this zone plays an important role in the individual identification of fish and the determination of its hierarchical rank [43,53]. All red coloration zones can play a determining role in defining a partner’s attractiveness [59,60]. In other fish species, in addition to red, orange and yellow often occur and perform the same function [61] but make the fish more visible to predators [62,63]. Therefore, the intensity of the coloration of the red zones is maintained at the same level regardless of size and age.

The results of the linear regression revealed a negative association between the BBCF zone and fish size and age, with the zone becoming darker. In *N. guentheri* male with typical coloration and size, the BBCF is black, which points to the presence of only one pigment, melanin, the content of which increases with age (Appendix A). Black areas of coloration on the fish body may also have important evolutionary relevance for protection against solar insolation and the amount of incoming ultraviolet light, the effects of which can be deleterious in shallow ephemeral pools [64].

The comparison of the two different age groups of individuals according to the studied parameters showed results similar to those of the linear regression analysis. In the first group of zones, no significant differences were observed between the age groups in terms of MoGVs and MeGVs, except in the BRIO zone, where the MeGVs between old and young individuals differed significantly. This suggests that the MeGV is preferable to the MoGV for detecting differences in coloration.

In turn, the zones of the second group showed more significant results. Thus, both studied zones (RSGC and BBCF) showed significant differences between old and young individuals in MoGV and MeGV parameters. Given the reliability of both parameters in determining coloration intensity, one parameter is not superior to the other in calculating gray values.

Overall, the results of the color assessment method used in this study were more reliable when comparing coloration intensity in two different age groups of individuals than when comparing mass samples. As a result, there was a gradual decrease in the level of coloration intensity to minimal values in the group of older males. As this study did not include individuals with visible morphological changes as a result of aging, no dramatic changes in coloration were observed.

In this work, the intensity of coloration zones determined by phenotypic mechanisms was measured, but some aspects of coloration changes observed in social interaction (e.g., establishing social hierarchy) and group content may not have been observed. Some fishes show extremely short-term changes in coloration as a result of group interaction between males and changes in their hierarchical status [65,66]. It was noted that in the genus *Nothobranchius*, the change in coloration intensity in response to individual status is slow, as it is mediated hormonally rather than neuronally [67].

Changes in the patterns of coloration of male *N. guentheri* with aging are associated with complex physiological and biochemical reactions related to hormonal changes and the level of expression of genes responsible for the differentiation of pigment cells [68,69]. Age-dependent metabolic disorders cause regressive changes in the balance of apoptosis and blastomelanophore differentiation [69].

Despite the diversity of coloration of the different body regions of adult male *N. guentheri*, we were able to demonstrate significant changes in the intensity of coloration of individual areas according to the age and size of the fish using new photofixation methods and image processing software. The methods developed in this study can be applied to both short-lived representatives of the family *Cyprinodontiformes* and to other model organisms. It can be assumed that for fish with more uniform coloration, the measurements of coloration intensity will be easier due to the reduction in the number of zones that need to be measured. In addition, this measurement method is suitable both for assessing individual properties of fish and for studying the coloration intensity of large groups of individuals, which may be required in toxicological studies, studies of the effect of pharmacological drugs, and studies of the expression of genes responsible for changes in coloration [70], microevolutionary processes of phenotype change [22] and male–male aggression acts to suppress reproduction physiologically [22,71,72,73].

Based on the data collected, the coloration of male *N. guentheri* is highly variable, even in genetically homogeneous individuals. As the methods used in the present study do not require high-tech, costly equipment, they can be recommended for use in studies aimed at estimating color changes, even minor changes, in fish species. As members of this fish species have a short duration of ontogenesis, coloration changes that occur during most of their life cycle were considered in the present study. As shown by our results, changes in the coloration of male *N. guentheri* during its life cycle were expressed by changes in both the color intensity and size of specific areas in particular zones.

## 5. Conclusions

As shown by our results, the methods proposed in this study are suitable for comparative analysis of the coloration of *N. guentheri* males, both in individual fish and among groups of fish of different ages and sizes. The assessment of changes in fish coloration revealed patterns of change in both the overall color intensity and coloration of individual zones. When the entire sample consisting of individuals of different sizes was assessed, the selected zones reliably distinguished between individuals of different age groups. As shown by the measurements of specific areas of the body in particular zones, there was a positive association between these measurements and fish age. When the fish were divided into two groups by age and size, in the zones in the first group, there was a significant difference in average gray values only in the BROI zone. The zones in the second group showed reliable differences when comparing the groups among themselves, as well as within the same group, depending on the method of measurement. Based on the results of this study, the measurement of gray modal and mean values can be applied to assess age-related changes in coloration of male *N. guentheri* and other small fish with pronounced individual coloration. The results confirm the possibility of *N. guentheri* as a model organism for studying the mechanisms of development and aging.

## Figures and Tables

**Figure 1 biology-11-00205-f001:**
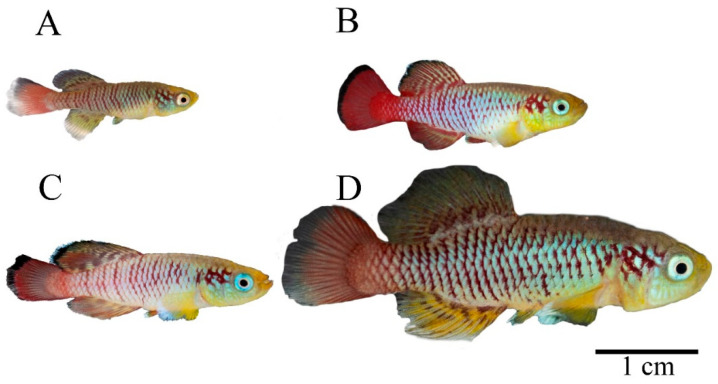
Morphological changes in the coloration of male N. guentheri, depending on hierarchical status and age. (**A**) Young male with signs of primary coloration (1.5 months); (**B**) Young male of high hierarchical status (4.5 months); (**C**) Adult male (6 months); (**D**) Adult male with signs of coloration regression (8 months).

**Figure 2 biology-11-00205-f002:**
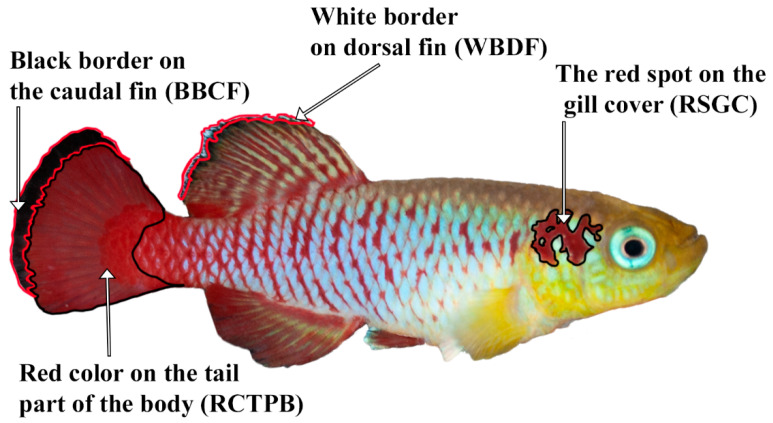
First set of body zones used in the measurement of *N. guentheri*.

**Figure 3 biology-11-00205-f003:**
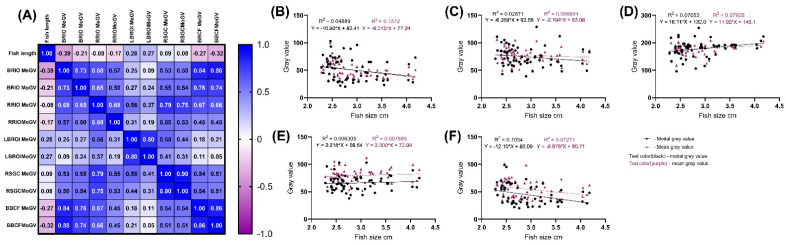
Linear regression of MoGV and MeGV parameters as a function of male *N. guentheri* size. (**A**) Person’s correlation matrix between different zone. (**B**) BROI linear regression; (**C**) LBRIO linear regression; (**D**) RROI linear regression; (**E**) BBCF linear regression; (**F**) RSGC linear regression.

**Figure 4 biology-11-00205-f004:**
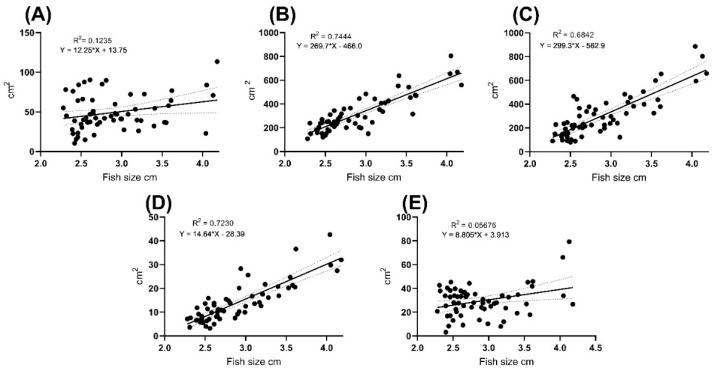
Linear regression of MoGV and MeGV parameters as a function of male *N. guentheri* size. (**A**) BROI linear regression; (**B**) LBRIO linear regression; (**C**) RROI linear regression; (**D**) BBCF linear regression; (**E**) RSGC linear regression.

**Figure 5 biology-11-00205-f005:**
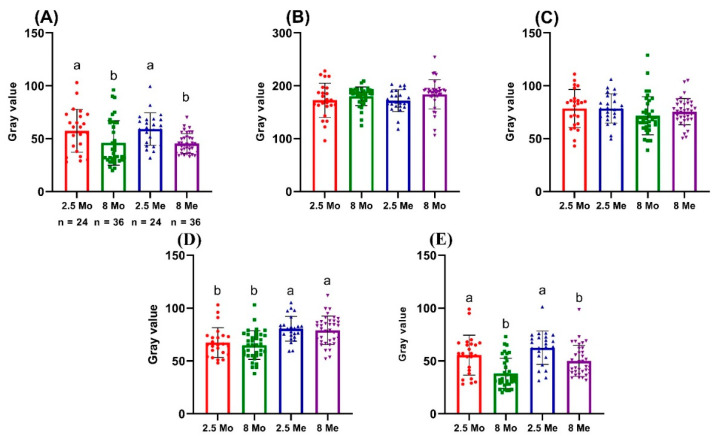
Comparison of MoGV and MeGV parameters in 2.5- and 8-month-old fish. The value (*p* < 0.05) from one-way ANOVA with comparison using Tukey’s post hoc analysis. Different superscript letters (a,b) indicate statistically significant differences between the experimental groups. (**A**) BROI zone; (**B**) LBRIO zone; (**C**) RROI zone; (**D**) RSGC zone; (**E**) BBCF zone.

**Figure 6 biology-11-00205-f006:**
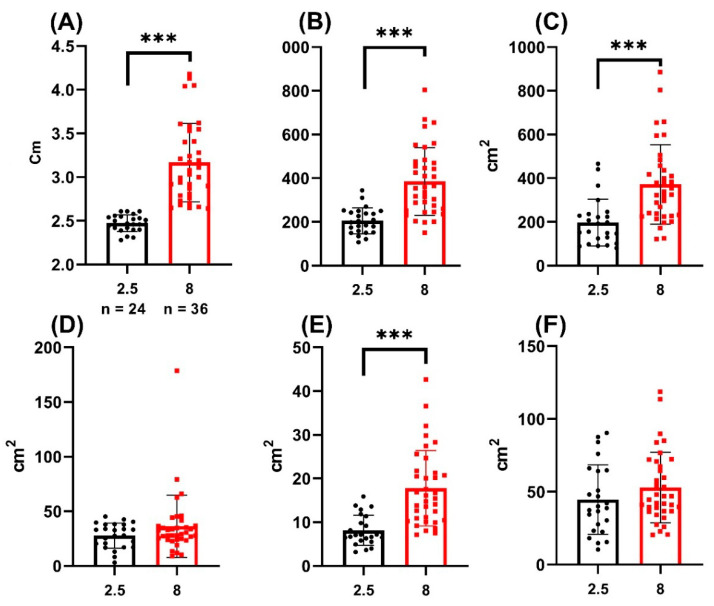
Comparison of the coloration areas of different zones of males of different age groups (2.5- and 8-month post-hatching). The value (*** *p* < 0.001) from one-way ANOVA with comparison using Tukey’s post hoc analysis. (**A**) fish length; (**B**) LBRIO zone; (**C**) RROI zone; (**D**) RSGC zone; (**E**) BROI zone; (**F**) BBCF zone.

**Figure 7 biology-11-00205-f007:**
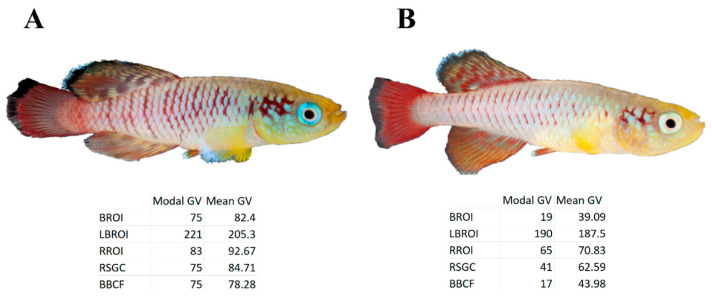
A comparison of the coloration of various male *N. guentheri* of comparable ages and sizes according to the gray value of the studied zones. (**A**) *N. guentheri* male with typical coloration and size. (**B**) Male with deviant type of coloration.

**Table 1 biology-11-00205-t001:** Numerical measurement data and the MoGVs for the two sets of zones in the entire sample. The value (*p* < 0.05) from unpaired *t*-test with Welch’s correction.

	8 MPH	2.5 MPH	*p*-Value	Whole Sample
Fish length	2.47 ± 0.09	3.16 ± 0.44	≤0.0001	2.89 ± 0.49
BROI MeGV	45.44 ± 9.42	59.11 ± 15.27	0.0145	50.91 ± 13.75
RROI MeGV	75.49 ± 12.45	78.62 ± 13.84	0.8733	76.74 ± 13
LBROI MeGV	183 ± 27.63	171.9 ± 20.66	0.2528	176.9 ± 19.16
BBCF MeGV	50.03 ± 14.65	62.62 ± 15.77	0.0153	55.06 ± 16.22
RSGC MeGV	78.98 ± 13.46	80.49 ± 11.72	0.9978	79.58 ± 12.71
BROI MoGV	46.06 ± 21.06	57.46 ± 20.23	0.9525	50.62 ± 21.31
RROI MoGV	71.72 ± 18	78.58 ± 17.96	0.5817	74.47 ± 18.15
LBROI MoGV	180.2 ± 17.62	172.5 ± 32.45	0.8143	178.5 ± 29.76
BBCF MoGV	38.19 ± 14.34	55.50 ± 18.92	0.0003	45.12 ± 18.29
RSGC MoGV	64.97 ± 13.56	67.42 ± 14.07	0.8563	65.95 ± 13.70

**Table 2 biology-11-00205-t002:** Numerical data on X and the results of what analysis of the zones in the two groups. The value (*p* < 0.05) from unpaired *t*-test with Welch’s correction.

	BROI Area	RROI Area	LBROI Area	RSGC Area	BBCF Area
8 MPH	52.95 ± 24.26	371.7 ± 182	385.1 ± 155.1	17.75 ± 8.62	36.38 ± 28.44
2.5 MPH	44.67 ± 23.9	197.4 ± 106.9	205.5 ± 59.21	8.13 ± 3.45	27.85 ± 11.42
*p*-value	0.1975	<0.0001	≤0.001	≤0.001	0.1686
Whole sample	49.64 ± 24.46	302 ± 177.5	313.3 ± 153.3	13.91 ± 8.44	32.97 ± 23.42

## Data Availability

Not applicable.

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
