# Peer review of "Evaluation of Age-Dependent Changes in the Coloration of Male Killifish Nothobranchius Guentheri Using New Photoprocessing Methods"

_biology, 2022, doi:10.3390/biology11020205_

Round 1

Reviewer 1 Report

The paper is well written and interesting. The authors should explain better the aim of the work and the impact/application of the obtained results in Notobranchius’s research.

Author Response

Dear reviewer

We are grateful to the Reviewer for the comments/suggestions that allowed us to improve our manuscript entitled: “Evaluation of age-dependent changes in the coloration of male killifish Nothobranchius guentheri using new photoprocessing methods”.

All the minor comments and suggestions were taken into consideration. When required and appropriate, the original text was modified using highlight mode. The authors modelled their replies using a C# for the reviewer questions and an R# to reply to each of the comments.

The paper is well written and interesting. The authors should explain better the aim of the work and the impact/application of the obtained results in Notobranchius’s research.

R- In our work we wanted to demonstrate the usability of Nothobranchius guentheri as a model for biological research. It was shown that the morphological features of these species can also serve as reliable markers of aging.

R- Thank you for the suggestions that will improve the manuscript. Adjusted the aim of the study (line 103-104);

R- Annotation expanded (line 43-44);

R- Conclusions expanded (line 426-427);

Reviewer 2 Report

I have read with great interest the work on this exciting emerging model in research. All the works concerning the biology of this animal arouse a curiosity since the particular condition in which they live and develop. Also in this case the paper  appears to be well-conceived and carried out, with the consistent conclusions based on the  observations produced. I only have a few small notes that I recommend to adjust to further increase the form of this work. Here is a short list of my exceptions:

 -  Line 54 I don’t quite understand the phrase, there is a “of” which I don’t know what you mean.

- in the captions of all figures the letters in brackets are in

lowercase while on the graph they are capitalized.Consider to uniform it.

-Line 264 correct BRIO in BROI.

Author Response

Dear reviewer

We are grateful to the Reviewer for the comments/suggestions that allowed us to improve our manuscript entitled: “Evaluation of age-dependent changes in the coloration of male killifish Nothobranchius guentheri using new photoprocessing methods”.

All the minor comments and suggestions were taken into consideration. When required and appropriate, the original text was modified using highlight mode. The authors modelled their replies using a C# for the reviewer questions and an R# to reply to each of the comments.

I have read with great interest the work on this exciting emerging model in research. All the works concerning the biology of this animal arouse a curiosity since the particular condition in which they live and develop. Also in this case the paper appears to be well-conceived and carried out, with the consistent conclusions based on the observations produced. I only have a few small notes that I recommend to adjust to further increase the form of this work. Here is a short list of my exceptions:

C1- Line 54 I don’t quite understand the phrase, there is a “of” which I don’t know what you mean.

R1- Thank you for the comment. This is a technical mistake, corrected.

C2- in the captions of all figures the letters in brackets are in lowercase while on the graph they are capitalized. Consider to uniform it.

R2- Thank you for the note. The captions have been corrected.

C3- Line 264 correct BRIO in BROI.

R3- Thank you for comment. This is a technical mistake, corrected (line 267).

Reviewer 3 Report

General comments:

A scientifically interesting article using new photoprocessing methods in the determination of the pattern of pigmentation of male killifish. The manuscript is well written and requires minor revision. Some topics should be specified. Proofreading by a native English speaker is recommended.

I want the authors to explain the difference between fish coloration/coloration pattern and fish pigmentation? Are they the same? Please discuss and clarify this more in deep.

The introduction needs to be improved, focusing on a few issues. I.e, It about aging, new images and methods, the onset of reproduction in fish, a new model of study,?

The discussion is too long, and please be more precise and concise. 

Specific comments:

LINE 29. It sounds vague to me “can be applied to other fish” which one and why? Please give more context.

LINE 52 and 68. Ie. Danio rerio and Oryzias latipes are in italic. 

LINE 52. Please give more context or rephrase this. “However…..with age” adding references.

LINE168-169. How have the authors selected those 60 images?

LINE 239, please change the order to table 1 and figure 3 in the text.

LINE 292, What other laboratories? 

LINE 423. Delete “in” 

LINE 445-449. Delete or change using the proper inform

Author Response

Dear reviewer

We are grateful to the Reviewer for the comments/suggestions that allowed us to improve our manuscript entitled: “Evaluation of age-dependent changes in the coloration of male killifish Nothobranchius guentheri using new photoprocessing methods”.

All the minor comments and suggestions were taken into consideration. When required and appropriate, the original text was modified using highlight mode. The authors modelled their replies using a C# for the reviewer questions and an R# to reply to each of the comments.

A scientifically interesting article using new photoprocessing methods in the determination of the pattern of pigmentation of male killifish. The manuscript is well written and requires minor revision. Some topics should be specified. Proofreading by a native English speaker is recommended.

I want the authors to explain the difference between fish coloration/coloration pattern and fish pigmentation? Are they the same? Please discuss and clarify this more in deep.

The introduction needs to be improved, focusing on a few issues. I.e, It about aging, new images and methods, the onset of reproduction in fish, a new model of study?

The discussion is too long, and please be more precise and concise.

R- Thank you for the suggestions. In our study, we used the term coloration as a set of morphological traits and manifestations under the influence of environmental factors, hierarchical status, and age. Pigmentation is the variety and number of different types (chromatophores) of pigmented cells in a particular area of the body.

R- Thank you for the suggestion. The structure of the introduction has been revised.

R- Thank you for comment. Changes have been made to the discussion section, and some sentences have been changed.

Manuscript comments:

C1- LINE 29. It sounds vague to me “can be applied to other fish” which one and why? Please give more context.

R1- Thank you, we agree with the suggestion, the following was changed (line 29-30).

C2- LINE 52 and 68. Ie. Danio rerio and Oryzias latipes are in italic.

R2- Thank you, corrections have been made throughout the text.

C3- LINE 52. Please give more context or rephrase this. “However…..with age” adding references.

R3 – Thank you for comment. The wording was checked (line 55-56).

C4- LINE168-169. How have the authors selected those 60 images?

R4- Thank you for comment. Photographs even with minimal defects and distortions were not used for analysis (line 170-171).

C5- LINE 239, please change the order to table 1 and figure 3 in the text.

R5- It was done.

C6- LINE 292, What other laboratories?

R6- This sentence was deleted during the revision of the discussion.

C7- LINE 423. Delete “in”

R7- Thank you for the comment. This is a technical mistake, corrected.

C8- LINE 445-449. Delete or change using the proper inform

R8- Thank you for comment. It was done